# Rv0687 a Putative Short-Chain Dehydrogenase Is Required for In Vitro and In Vivo Survival of *Mycobacterium tuberculosis*

**DOI:** 10.3390/ijms25147862

**Published:** 2024-07-18

**Authors:** Gunapati Bhargavi, Mohan Krishna Mallakuntla, Deepa Kale, Sangeeta Tiwari

**Affiliations:** Department of Biological Sciences, Border Biomedical Research Centre, University of Texas at El Paso, El Paso, TX 79968, USA

**Keywords:** *Mycobacterium tuberculosis* pathogenesis, oxidative stress, nitrosative stress, short-chain dehydrogenase (SDR), macrophage proinflammatory cytokines, macrophage-*Mtb* infection, mice-*Mtb* infection

## Abstract

*Mycobacterium tuberculosis* (*Mtb*), a successful human pathogen, resides in host sentinel cells and combats the stressful intracellular environment induced by reactive oxygen and nitrogen species during infection. *Mtb* employs several evasion mechanisms in the face of the host as a survival strategy, including detoxifying enzymes as short-chain dehydrogenases/reductases (SDRs) to withstand host-generated insults. In this study, using specialized transduction, we have generated a Rv0687 deletion mutant and its complemented strain and investigated the functional role of Rv0687, a member of SDRs family genes in *Mtb* pathogenesis. A wildtype (WT) and a mutant *Mtb* strain lacking Rv0687 (RvΔ0687) were tested for the in vitro stress response and in vivo survival in macrophages and mice models of infection. The study demonstrates that the deletion of Rv0687 elevated the sensitivity of *Mtb* to oxidative and nitrosative stress-inducing agents. Furthermore, the lack of Rv0687 compromised the survival of *Mtb* in primary bone marrow macrophages and led to an increase in the levels of the secreted proinflammatory cytokines TNF-α and MIP-1α. Interestingly, the growth of WT and RvΔ0687 was similar in the lungs of infected immunocompromised mice; however, a significant reduction in RvΔ0687 growth was observed in the spleen of immunocompromised Rag^−/−^ mice at 4 weeks post-infection. Moreover, Rag^−/−^ mice infected with RvΔ0687 survived longer compared to those infected with the WT *Mtb* strain. Additionally, we observed a significant reduction in the bacterial burden in the spleens and lungs of immunocompetent C57BL/6 mice infected with RvΔ0687 compared to those infected with complemented and WT *Mtb* strains. Collectively, this study reveals that Rv0687 plays a role in *Mtb* pathogenesis.

## 1. Introduction

*Mycobacterium tuberculosis* (*Mtb*) is the causative agent of tuberculosis (TB), a bacterial disease that claimed about 1.5 million lives worldwide in 2022 [1]. On infection, *Mtb* is phagocytosed by the sentinel cells of the host, macrophages. These cells are equipped with cell autonomous innate immune responses, which act as front-line defenders against *Mtb* infection [2,3]. Activated macrophages upregulate various antimicrobial pathways including the expression of inducible nitric oxide synthase (INOS), phagocyte NADPH oxidase (NOX2/gp91^phox^) and xanthine oxidase system that becomes recruited to the *Mtb* phagosome and generates antimicrobial reactive nitrogen and oxygen species (ROS and RNS) via a respiratory burst to kill *Mtb* [3,4,5,6]. Mice deficient in both *phox* and *iNOS* are highly susceptible to *Mtb* infection, which shows that RNS and ROS generation is required by the host for protection against *Mtb* [7]. These molecules alter the intracellular signaling events, triggering the host antimicrobial responses that primarily aim to eliminate *Mtb* [2,8]. The host cell ROS and RNS interacts directly with its bacterial targets and during the process, becomes converted into chemically district oxidants such as hydrogen peroxide (H_2_O_2_), hypochlorites (HClO), peroxynitrite (ONOO^−^) and hydroxy radicals [9,10]. These reactive species damage *Mtb* DNA, oxidize lipids and proteins, including highly sensitive proteins such as iron–sulfur (4Fe-4S) cluster proteins [9]. Thus, *Mtb* infection prompts macrophages to generate ROS and RNS as a mechanism to kill intracellular bacteria [10,11]. The ROS and RNS production by *Mtb*-infected macrophages, as part of host cell-autonomous innate immune response, not only play a role in direct host defense against the bacteria, but also trigger signaling pathways that contribute to the overall antimicrobial responses of these immune cells [12].

The survival of *Mtb* in the host not only depends on the manipulation of the host but also on its ability to encounter stresses imposed by immunologically active macrophages. *Mtb* employs several detoxifying antioxidant thiols and enzyme systems, such as superoxide dismutase (SOD), catalase-peroxidase, alkylhydroperoxidase (AhpC), and thioredoxin reductase [9,13]. However, the gene encoding global stress regulator *oxyR*, commonly found in several other bacteria, is absent in *Mtb*. The absence of *oxyR* in *Mtb* is suggested to be compensated by the presence of several AhpC, which help the bacteria tolerate ROS and RNS induced by the infected host cells [14,15]. In addition to AhpC, recent research has revealed that *Mtb* possesses a category of enzymes known as short-chain dehydrogenase/reductases (SDRs), which aid in managing peroxidase stress, in vitro detoxification as well as in vivo survival of *Mtb*, drug resistance and overall homeostasis [16,17]. The SDRs are one of the largest protein families, distributed widely in several organisms and classified into seven families such as classical, atypical, extended, intermediate, divergent, complex and unassigned [18,19]. SDRs can utilize a range of diverse substrates, such as alcohols, sugars and aromatic compounds [19]. SDRs represent various enzyme classes such as isomerases, lyases and oxidoreductases, although the majority of the SDRs are oxidoreductases [20,21]. These SDRs share an NAD(P)^+^ Rossmann-fold domain for substrate binding and use NADH or NADPH as its electron donors [21]. Moreover, SDRs are involved in metabolic pathways such as detoxification, drug resistance, steroid metabolism, and biosynthetic pathways of bacteria [17,22]. In the context of *Mtb* infection, specific SDRs have been implicated in modulating the activation or inactivation of certain drugs used in TB treatment, highlighting the involvement of SDRs in bacterial physiology and survival [23,24]. The genome of *Mtb* possess 23 SDRs, namely Rv0148, Rv0484c, Rv0687, Rv0851c, Rv0927c, Rv0945, Rv1144, Rv1245c, Rv1865c, Rv1882c, Rv1928c, Rv1941, Rv2214c, Rv2509, Rv2766c, Rv2857c, Rv3085, Rv3174, Rv3224, Rv3485c, Rv3502c, Rv3548c and Rv3549c (as listed in Mycobrowser) and only a few of these are characterized [25,26].

Strikingly, a transposon mutant of Rv0687, SDR has also shown to be attenuated for growth in both mice and primate models using the DeADMAn approach (designer arrays for defined mutant analysis) [26,27]. Though, the DeADMAn approach has limitations such as the polar effect due to the inactivation of downstream genes in operon. Therefore, we investigated the role of Rv0687 in Mtb pathogenesis and survival. To avoid any polar effects that leads to inactivation of downstream genes, this study involves generating precise knockout in-frame deletions using the strategy described by Jain et al. 2014 [28,29]. Rv0687 has homologs in other mycobacteria that are characterized as mycofactocin-associated dehydrogenases with non-exchangeable NAD-cofactors and as a putative carveol dehydrogenase [30]. Notably, Rv0687 is a non-essential in the *Mtb* genome, with a gene length of 828 base pairs, encoding a 275 amino acid protein with a molecular mass of 28 kDa. In the current study, we generated a knockout mutant of Rv0687 (RvΔ0687) using specialized transduction and characterized the phenotype of the mutant in comparison with the wildtype (WT) and complemented *Mtb* strains (C-Rv0687). Our results demonstrate that *Mtb* requires Rv0687 in vitro to cope up with the oxidative stress, nitrosative stress and for survival as well as growth in the mouse bone marrow-derived macrophage (BMDM) infection assays. Moreover, it is required by *Mtb* for survival and proliferation in vivo in immunocompetent and immunocompromised mice models of infection.

## 2. Results

### 2.1. Growth Kinetics of RvΔ0687 in Broth Media

RvΔ0687 deletion strain and its complemented strains were generated and confirmed (Appendix A) as mentioned in the Section 4. To assess the role of Rv0687 in *Mtb* survival WT, RvΔ0687 and C-Rv0687 strains were grown in both 7H9 complete and 7H9 dextrose media lacking OADC, and their survival was monitored via CFU assay for 12 days. In 7H9 complete media, the survival of RvΔ0687 showed no significant differences compared to the WT and C-Rv0687 at any of the tested time points (Figure 1A). Whereas in 7H9-dextrose media, all the strains were able to sustain without any change in survival until day 4, followed by a gradual decline in CFUs, which is not statistically significant at day 8. However, no CFUs were obtained for the RvΔ0687 at day 12, whereas WT and C-Rv0687 were able to persist in the 7H9 dextrose media and form CFUs (Figure 1B). The 7H9 dextrose media showed the difference in growth response among the *Mtb* strains.

### 2.2. RvΔ0687 Is Sensitive to ROS and Nitrite Stress Generating Agents

As we have seen differences in growth response among the *Mtb* strains in 7H9 dextrose media, we evaluated the response of WT, RvΔ0687 and C-Rv0687 to in vitro ROS using H_2_O_2_ and tBooth. In our pilot experiment (Appendix A), we used 7H9 complete media supplemented with glycerol, tyloxapol and OADC, treated with 5 mM H_2_O_2_ to perform the stress response studies, and we did not observe significant growth defect between the WT, RvΔ0687 and C-Rv0687 strains. However, the RvΔ0687 showed sensitivity to exposure of 5 mM and 10 mM H_2_O_2_ as determined by the reduction in CFUs, compared to the WT and C-Rv0687 in 7H9 dextrose media lacking catalase. At 5 mM H_2_O_2_ exposure, a significant reduction in RvΔ0687 CFUs was observed only at 96 h, whereas at 10 mM H_2_O_2_ exposure, at 24 h, a decrease in CFUs was observed, but at 48 h, the CFUs of RvΔ0687 dropped by 2 logs compared to WT and C-Rv0687 strains. Furthermore, RvΔ0687 was so susceptible to H_2_O_2_ stress that we recovered no CFUs at 96 h post-treatment compared to the WT and C-Rv0687 (Figure 2A,B). In addition, the RvΔ0687 was sensitive to 2 mM tBOOH at 24 h, with significantly declined CFUs compared to WT and the C-Rv0687 strains. However, none of the strains were able to sustain their growth after 48 h, as observed by the lack of CFUs in the presence of 2 mM tBOOH (Figure 2C). The nitrite stress response was assessed using 2 mM SNAP and 2 or 10 mM SN. At 72 and 96 h post-treatment, we observed a log decrease in the RvΔ0687 compared to WT and the C-Rv0687 strains in response to 2 mM SNAP treatment (Figure 2D). However, neither of the tested bacterial strains (WT, RvΔ0687 and C-Rv0687) showed any difference in CFUs on treatment with 2 or 10 mM SN (Appendix A) or in response to metal ions (copper, zinc, and iron) stress at any of the tested time points after post-treatment (Appendix A).

### 2.3. Deletion of Rv0687 Gene Does Not Impact the Mtb NADH, and NAD^+^ and NADH/NAD^+^ Ratio In Vitro

The NADH and NAD^+^ ratios have a crucial role in the redox homeostasis of a bacterial cell. Consequently, we measured the concentrations of NADH (Figure 3A) and NAD^+^ (Figure 3B) as well as the NADH/NAD^+^ ratio (Figure 3C) for the WT, RvΔ0687 and C-Rv0687 strains. Based on our observations, it is evident that the deletion of Rv0687 has no significant impact on modulating the NADH or NAD^+^ levels and redox homeostasis of *Mtb* in aerobic conditions.

The WT, RvΔ0687 and C-Rv0687 were grown to 10^7^ bacteria and the cells were pelleted and lysed using NADH extraction buffer. NAD/NADH were extracted using 10 kD spin column and the resulting concentrations (pmol) were estimated at 1, 2, 3 and 4 h by measuring the absorbance at 450 nm after addition of NADH developer. The change in the spectral readings at various timepoints probably represents reactivity of proteins formation of measurable product due to interaction between NADH, NAD^+^ and NADH developer. Based on the readings, the concentration of NADH and NAD^+^ were estimated. Concentration of NAD^+^ was calculated by subtracting NADH from NAD^+^ (represents total conc of NAD^+^ and NADH). In addition, ratio of NADH/NAD^+^ is calculated using the formula NADH/NAD^+^. The experiment was performed in triplicates with biological samples.

### 2.4. Deletion of Rv0687 Enhances Susceptibility of Mtb to Antimycobacterial Drugs Delamanid and NMR711

To assess the sensitivity of WT, RvΔ0687 mutant and C-Rv0687 strains to Isoniazid (INH), Delamanid (DEL), Rifampicin (RIF), Bedaquiline (BDA) and NMR711, an Alamar Blue assay was performed. The RvΔ0687 displayed increased susceptibility to DEL at concentrations of 0.062–1 mg/L, which is lower than the survival rates observed for the WT and the C-Rv0687 strains (Figure 4A). In our previously published studies, we discovered that small compound NMR711 has anti-mycobacterial activity against Mtb at a concentration of 50 mg/L [31]. Interestingly, we observed that RvΔ0687 also displayed reduced survival for NMR711 at tested concentrations of 0.39–50 mg/L compared to WT and the C-Rv0687 strains (Figure 4B). Conversely, RvΔ0687 exhibited similar susceptibility to antimycobacterial drugs BDA and RIF as the WT and C-Rv0687 strains (Figure 4C–E).

### 2.5. RvΔ0687 Is Required for Intracellular Replication and Suppression of Early Inflammatory Immune Response in Macrophages

The in vitro survival of WT, RvΔ0687 and C-Rv0687 strains were tested using mouse BMDMs. As shown in (Figure 5A), no significant difference was observed in the uptake of the WT, RvΔ0687 and C-Rv0687 strains by BMDMs at day 0. Interestingly at day 1, though, we observed small differences in the CFUs between WT and RvΔ0687 strains, but the differences were significant at later stages at day 3, 5 and 7 post infections (Figure 5A). We found that *Rv0687* is required for the intracellular survival of *Mtb*, as the deletion mutant of *Rv0687* is attenuated for survival in the macrophages. It was noted that there was a slight change in the survival of C-Rv0687 compared to WT at 5 and 7 days post-infection, which is not statistically significant. We further assessed the production of RNS (Griess assay) and ROS (NBT assay) by BMDMs in response to infection with WT, RvΔ0687, and C-Rv0687 strains. Though there was a significant change in RNS and ROS production by WT-infected vs. uninfected BMDMs, we did not observe any significant change in the production of ROS or RNS by BMDMs during infection with either WT, RvΔ0687, and C-Rv0687 strains (Appendix A).

Furthermore, we measured the cytokines to determine the activation status of the macrophages. The expression levels of TNF-α, IL-1β, IP-10 and MIP-1α cytokines were analyzed in the WT, RvΔ0687 and C-Rv0687 strains in the cell-free post-infected supernatants of infected BMDMs at 1–5 days post-infection. Within 3 h of infection, compared to uninfected, WT-infected BMDMs showed 5-fold increased secretion of TNF-a, IL-1β and ~10-fold increase in MIP-1a and IP-10. Interestingly, RvΔ0687-infected BMDMs showed a significantly higher expression of TNF-α and MIP-1α at d1 and d5 days compared to WT- and C-Rv0687-infected BMDMs (Figure 5B,C). We have not observed any significant change in the secretion of IL-1β and IP-10 by RvΔ0687-infected BMDMs compared to that of WT- and C-Rv0687-infected BMDMs (Figure 5D,E).

### 2.6. RvΔ0687 Is Attenuated In Vivo in Immunocompromised and Immunocompetent Mice

The in vivo survival of WT, RvΔ0687 and C-Rv0687 strains were analyzed in immunocompromised (Rag^−/−^) and immunocompetent (C57BL/6) mice (Figure 6A). Rag^−/−^ mice were infected via aerosol with a dose of 200–250 CFU/mice. Mice were sacrificed at day 0, W4 and W8 to determine the CFU burden in the lungs and spleen. A group of mice infected with each strain were kept for survival studies. At W4 and W8 post-infection, WT, RvΔ0687 and C-Rv0687 showed similar bacterial burden in the lungs of Rag^−/−^ mice (Figure 6B), but in spleen at W4 the growth of RvΔ0687 was 8-fold lower than WT and C-Rv0687 strains (Figure 6C). Furthermore, we compared the survival of Rag^−/−^ mice after infection with WT, C-Rv0687 or RvΔ0687 strains. Interestingly, the WT or C-Rv0687 infected group started to die between 60 and 70 days post-infection, while the RvΔ0687 infected animals survived until 125 days post-infection (Figure 6D). This suggests that RvΔ0687 is impaired in virulence and plays a significant role in the in vivo survival of *Mtb* in Rag^−/−^ mice.

To further evaluate the role of Rv0687 in *Mtb* pathogenesis in vivo, we infected C57BL/6 mice and euthanized them at 4 and 8 weeks post-infection. Lungs and spleen were harvested, plated and CFUs were enumerated. In lungs, though at W4 CFUs were similar for WT, RvΔ0687 and C-Rv0687 strains, differences were observed during late infection at W8 and enumerated CFUs for RvΔ0687 were 3-fold lower than WT or C-Rv0687 (Figure 6E). Furthermore, in spleen, compared to WT or C-Rv0687 infected mice at W4 and W8, RvΔ0687-infected mice displayed ~4-fold or 2.5-fold lower CFU’s, respectively (Figure 6F).

## 3. Discussion

In this study, we investigated the functional role of *Mtb* Rv0687 that codes for a putative short-chain dehydrogenase. In bacteria including *Mtb*, it is reported that short-chain dehydrogenase/reductases (SDRs) aid in managing peroxidase stress, detoxification, in vitro, in vivo survival, drug resistance and overall homeostasis [16,17]. Better understanding of role of these SDR’s in *Mtb* pathogenesis will lead to the development of drugs and therapeutics in future to combat tuberculosis (TB). To understand the functional role of *Mtb* Rv0687, we created a gene knockout mutant of Rv0687 (RvΔ0687), its complemented strain (C-Rv0687) and characterized the role of Rv0687 in *Mtb* survival under various stress conditions including, oxidative, nitrite and drug-mediated stress. We also investigated the function of Rv0687 in *Mtb* infection in BMDMs and mice models. Overall, our study suggests that Rv0687 has a substantial role in *Mtb* pathogenesis in combating oxidative and nitrite stress, drug tolerance and host immunity and is indispensable for pathogenicity of *Mtb* in macrophages and mice models of infection.

Bacterial growth and survival are influenced by the composition of the growth media [32]. The OADC supplement is commonly added to *Mtb* growth media. Since *Mtb* growth can produce toxic by-products, such as ROS through metabolic processes, the presence of OADC (catalase) aids in neutralizing these toxic compounds by catalyzing the decomposition of hydrogen peroxide. This enzymatic activity supports better bacterial growth in OADC-containing media [33]. While performing survival kinetics, we observed that in nutrient-rich conditions and in the presence of catalase, the growth of RvΔ0687 was indistinguishable from that of WT and C-Rv0687 under in vitro conditions. We further used 7H9 media supplemented with glycerol, tyloxapol and 0.2% dextrose (7H9-dextrose) and found that WT, RvΔ0687 and C-Rv0687 strains were able to survive normally until day 8 and declined gradually until day 12, suggesting the susceptibility of the strain to media lacking the catalase and BSA [34,35,36]. Taken together, our findings highlight the contribution of the Rv0687 gene in the adaptation of *Mtb* to grow under stress conditions.

Since tolerance to ROS and nitrite stress plays an essential role in bacterial pathogenesis, we further investigated the response of *Mtb* strains against stress-generating agents [37]. In our pilot experiment (Appendix A), we used 7H9 complete media supplemented with glycerol, tyloxapol and OADC, treated with 5 mM H_2_O_2_ to perform the stress response studies, and we did not observe significant growth defect between the WT, RvΔ0687 and C-Rv0687 strains. In contrast, when we tested the phenotype of the *Mtb* strains in 7H9 dextrose media lacking OADC, we found that the RvΔ0687 was more sensitive to ROS-generating agents, as evidenced by a decline in its growth compared to WT and C-Rv0687 [38]. To conclude, the failure of RvΔ0687 growth might be due to enhanced oxidative stress mediated due to the absence of catalase for bacterial survival. Our results showed that RvΔ0687 is more sensitive to ROS (H_2_O_2_ and tBOOH). Our results are further supported by previous microarray studies that shows that Rv0687 is upregulated on *Mtb* treatment with H_2_O_2_ [39]. Like *Mtb sigJ, mel2* and *sod* that are known to play a role in providing resistance against ROS and *Mtb* pathogenesis [38,40,41], Rv0687 might also help *Mtb* in coping up the ROS and is required for *Mtb* pathogenesis.

In addition, we investigated the response of RvΔ0687 to nitrite stress by treating bacterial cultures with SNAP and SN. The increased sensitivity of RvΔ0687 to SNAP treatment suggests that the Rv0687 gene may play a role in *Mtb* defense against nitrosative stress induced by SNAP, which is a nitric oxide donor, as has been shown for the gene knockout mutant of Rv2617c of *Mtb* involved in virulence [42]. Our findings also suggest that RvΔ0687 can tolerate exposure to SN up to 10 mM concentrations; this was like the findings of a study by Suwarna et al., which showed that *Mtb* can resist and survive in the presence of a nitrite concentration up to 10 mM in broth media [43]. Furthermore, other possible explanation is that we have not observed any effect of sodium nitrite on the viability of the bacterial strains as we used sodium nitrite at physiological pH, as a donor of reactive nitrogen species (RNS), and it dissociates into reactive nitrogen species at acidic pH [44,45]; therefore, the acidification of growth media prior to NaNO2 addition would have been better for RNS studies. Additionally, we tested the response of RvΔ0687 to exposure to transitional metals such as copper, zinc and iron, which can induce oxidation reactions. However, we could not find a significant difference in the survival between the RvΔ0687 and WT or complemented strains. The *Mtb*, being an intracellular pathogen, grows inside the phagosomes of macrophages, where oxygen and nitrogen intermediates are produced; having such evasion mechanisms would enable *Mtb* to survive under stress conditions. SDRs in *Mtb* are involved in regulating peroxidase stress and detoxification. Interestingly, a homolog of Rv0687 in other mycobacterial species is characterized as a mycofactocin-associated dehydrogenase with non-exchangeable NAD-cofactors [46]. The homolog of Rv0687 in *M. paratuberculosis* is a putative carveol dehydrogenase, and its molecular structure has been reported. Accordingly, carveol has been shown to have an antioxidant role [47]. This suggests that Rv0687 might also be involved in redox reactions or synthesis of cellular metabolites that can play an important role in the detoxification of ROS or RNS in *Mtb* and can be further explored in the future. Therefore, understanding the specific mechanisms by which the Rv0687 gene confers protection against ROS and RNS could provide valuable insights into the bacterial stress response relating to the pathogenesis of *Mtb*.

We also investigated the impact of Rv0687 on the redox homeostasis of *Mtb*. In general, redox homeostasis, involving the balance between reduced NADH and oxidized NAD^+^, plays a key role in maintaining cellular redox equilibrium, which is crucial for bacterial survival and adaptation to various environmental stresses, including oxidative and nitrosative stresses. [48,49,50,51]. Hence, any disruption in redox homeostasis can have a significant impact on bacterial physiology and survival [52,53]. We found that the NADH and NAD^+^ concentrations, as well as the NADH/NAD^+^ ratio, were comparable between RvΔ0687 and the WT and C-Rv0687 strains. These results were consistent with the phenotype of the DosR regulon mutant of *Mtb* known to assist *Mtb* in metabolic homeostasis and recovery from dormancy, but the DosR mutant of *Mtb* exhibited similar NADH and NAD^+^ levels as the WT during aerobic growth conditions at early time points. There is a possibility of plasticity in SDRs that can compensate for the function of Rv0687, as it has been shown that deletion of a single succinate dehydrogenase (sdhA) gene, which is an SDR, does not have major redox potential in *Mtb* [48,52]. These observations suggest that Rv0687 might not be directly involved in regulating the balance between NADH and NAD^+^ in *Mtb* under the conditions tested. This could be due to the presence of redundant or alternative SDR enzymes that ensure the stability of the NADH/NAD^+^ ratio in the absence of RvΔ0687 under the tested conditions. Further research is required to elucidate the precise conditions in which Rv0687 might play direct or indirect role in balancing NADH/NAD^+^ ratio by interacting with other redox-related components of *Mtb*.

Rv0148 is an SDR of *Mtb* and the deletion mutant of Rv0148 has been associated with the sensitivity to antibiotics [16]. Therefore, we assessed the sensitivity of WT, RvΔ0687 and C-Rv0687 strains against anti-*Mtb* antibiotics, using a Alamar Blue assay. The results demonstrated distinct drug susceptibility patterns for the RvΔ0687 mutant. Notably, compared to the WT and C-Rv0687 strains, the RvΔ0687 showed increased susceptibility to two drugs: NMR711 and DEL. Conversely, the RvΔ0687 strain exhibited no significantly different sensitivity than the WT and C-Rv0687 strains against INH probably due to interference of Alamar Blue with INH [54]. We also did not observe significant differences in bacterial survival between WT and C-Rv0687 strains against BDA and RIF. This indicates that the Rv0687 plays a role in bacterial susceptibility towards antibiotics, particularly to DEL and NMR711 [55]. It correlates with our unpublished data that Rv0687 was upregulated in *Mtb* samples treated with NMR711. There is possibility that it might work synergistically with delamanid, a recently approved TB drug but for which additional studies are required. Further studies are also needed to elucidate the underlying mechanisms by which Rv0687 influences the bacterial susceptibility to these drugs.

During *Mtb* infection, the recognition of the pathogen by macrophages induces changes in the expression of cytokines that impact the elimination or progression of infection [56]. To assess the infection and in vivo survival of WT, RvΔ0687 and C-Rv0687 strains, we used a mouse BMDMs infection model and monitored the intracellular bacterial survival after infection. We observed that RvΔ0687 was defective for intracellular survival compared to WT and C-Rv0687 strains in BMDMs. The current results were consistent with the findings in another SDR and Rv2159c alkylhydroperoxidase of *Mtb*. The Rv0148 and Rv2159 mutants were defective for intracellular survival, although it induced proinflammatory cytokines in a macrophage model [15,16]. Differential survival of C-Rv0687 strain might be attributed to increased hypoxia conditions and the use of mycobacterial vector pMV361 for complementing the Rv0687 gene. Compared to the uninfected, we have observed significant changes in the production of ROS and RNS by infected BMDMs though there were no significant changes among WT vs. RvΔ0687 strains.

Further, *Mtb*-infected phagocytes generate ROS and RNS, which is required to control or kill Mtb in infected macrophages. Other than its antimicrobial effect, ROS and RNS generated by macrophages have been shown to influence the activation of T cell responses, production of proinflammatory cytokines, induce autophagy, and apoptosis [10]. Due to the change in the ROS and RNS production during BMDMs infection with Mtb and RvΔ0687, we further hypothesized this could impact the secretion of cytokines or chemokines and analyzed the levels of IL-1β, TNF-α, MIP-1α and IP-10 cytokines in the supernatants of BMDMs infected with *Mtb* strains. Strikingly, we observed that RvΔ0687-infected BMDMs showed significantly higher TNF-α and MIP-1α levels compared to both WT and C-Rv0687 strains. MIP-1α/CCL3 is a chemotactic chemokine secreted by macrophages to recruit inflammatory cells and maintain effector immune response [57]. This suggests that the Rv0687 gene might influence the host immune response during *Mtb* infection in macrophages.

To investigate the in vivo survival and virulence of RvΔ0687, in comparison to the WT and C-Rv0687 strains, we used two mice models of pulmonary infection. Rag^−/−^ mice, which lack functional T and B cells, is considered to be an ideal model to screen the mutant survival as they will be more vulnerable to infection compared to C57BL/6 mice. In Rag^−/−^ mice, all the tested *Mtb* strains exhibited similar bacterial burden in the lungs at 4 weeks. However, in the spleen, RvΔ0687 displayed a significantly lower bacterial growth compared to WT and C-Rv0687 at this time. This suggests that Rv0687 may play a role in the dissemination of *Mtb* primary infection from lung to spleen and early establishment of *Mtb* infection in the spleen of immunocompromised mice. But at 8 weeks post-infection, all the strains showed a comparable bacterial burden in both the lungs and spleen, indicating that the RvΔ0687 mutant was able to overcome the initial difficulty in spleen colonization. In addition, RvΔ0687 mutant-infected Rag^−/−^ mice survived for a longer duration than the WT-infected mice. There is a possibility that RvΔ0687 might induce lesser lung inflammation and pathology compared to WT, leading to prolonged survival. Furthermore, there is a possibility that there is difference in the natural killer (NK)-mediated response generated by WT vs. RvΔ0687. Interestingly, a study by Feng et al. showed the contribution of interferon-gamma secreting natural killer (NK) in the survival of Rag^−/−^ mice after *Mtb* infection in the absence of T and B cells compared to p40^−/−^Rag^−/−^ mice [58]. However, additional detailed studies are needed to test this hypothesis.

Interestingly, in C57BL/6 mice, a similar bacterial load was observed in the lungs of WT, RvΔ0687 and C-Rv0687 strains after 4 weeks of infection. However, at 8 weeks post-infection, RvΔ0687 displayed a 3-fold lower bacterial burden in the lungs and ~3–4 fold reduced bacterial load in the spleen at both 4 and 8 weeks post-infection. As the CFU of WT and RvΔ0687 were similar in the lungs at 4 weeks (an initial time point) we do not expect tremendous differences in pathology of lungs by H&E staining; however, these can be analyzed in our future studies. The observed phenotype for RvΔ0687 at 8 W post infection but not earlier, suggests that possibly Rv0687 gene is required by Mtb to counteract the host adaptive immunity during the chronic stages of infection (i.e., 8 W post infection). Another possibility is the differential adaptive response generated by host in response to infection with RvΔ0687 KO mutant compared to WT *Mtb* to counteract the host response. Together, these findings shed light on the complex interplay between *Mtb* and the host immune system, emphasizing the significance of the Rv0687 in modulating *Mtb* survival in mouse lungs and spleen. Further studies involving biochemical characterization of the SDRs reported in this study and its interaction with other regulatory networks will shed light on its role in bacterial adaptation to the host environment, tolerance to antibiotics and in *Mtb* pathogenesis.

## 4. Materials and Methods

### 4.1. Bacterial Strains

Wildtype *Mtb* bacterial strains and vector plasmids (p004s, phAE159, pMV361) used in this study were gifted by Dr. William R Jacob Jr (Albert Einstein College of Medicine) and are listed in Appendix A. All the Mtb strains were grown at 37 °C in Middlebrook 7H9 (Difco, Burlington, MA, USA) media, supplemented with 10% OADC (oleic acid-albumin-dextrose-catalase), 0.5% glycerol and 0.05% tyloxapol designated in the text as 7H9 complete media. Wherever required, *Mtb* strains were grown in 7H9 media supplemented with dextrose (0.2%), 0.5% glycerol and 0.05% tyloxapol (referred as 7H9 dextrose media) for in vitro ROS and RNS sensitivity studies. To grow *Mtb* strains carrying antibiotic markers, hygromycin (75 mg/mL) or kanamycin (20 mg/mL) were supplemented to the media wherever needed.

### 4.2. Gene Knockout Construction and Confirmation

The Mtb genomic DNA was extracted using CTAB-NaCl method [59]. Gene knockout of *Mtb* (CDC1551) was performed by specialized transduction as described previously [28]. Briefly, the flanking regions of the *Rv0687* gene sequence were obtained from Tuberculist and cloned into the p0004-SacB vector, carrying the Hyg^R^ resistance gene for homologous recombination. The recombinant construct was further cloned into phAE159 temperature-sensitive phasmid and a knockout mutant of *Rv0687* was generated using specialized transduction. Deletion of *Rv0687* was confirmed using 3-primer strategy [29] (Appendix A), and whole genome sequencing was performed on a MiSeq instrument (Illumina, San Diego, CA, USA) following the protocol provided by Illumina [60]. Further, the mutant strain was complemented by integrating the wildtype copy of the gene into the genome of the RvΔ0687 mutant using a pMV361 mycobacterial integrative vector and complementation was confirmed using gene-specific primers (Appendix A) and kanamycin primers [29]. The details of the strains, plasmids, phasmids and primers used in the study are provided in Appendix A.

### 4.3. Assessing Mtb Response to ROS and Nitrite Stress Generating Agents

The wildtype, RvΔ0687 and C-Rv0687 *Mtb* strains were grown in 7H9 complete media to OD_600 nm_ ≈ 0.8–1. Bacterial strains were centrifuged and pellet was washed with PBST to remove the catalase and diluted to 0.05 × 10^7^ cells/mL with 7H9 dextrose media to a final volume of 5 mL. Diluted bacterial cultures were treated with ROS; 5 or 10 mM hydrogen peroxide (H_2_O_2_) and 2 mM tert-butyl hydroperoxide (tBOOH) or nitrite stress; 2 mM S-nitroso-N-acetyl-D, L-penicillamine (SNAP) and 2 and 10 mM sodium nitrite (SN) generating agents; and the cultures were incubated for 24, 48, 72 and 96 h after post-treatment and the colony forming units (CFUs) were determined by plating the serially diluted bacterial culture onto 7H10 plates and incubating the plates at 37 °C for 4–6 weeks.

### 4.4. Quantification of Mtb NADH and NAD^+^ Levels

To quantify the NADH and NAD^+^ levels, we used the calorimetric kit from abcam (Cat no. ab65348) and protocol was followed as recommended by manufacturer. The wildtype, RvΔ0687 and C-Rv0687 bacterial strains were grown in 7H9 complete media to OD_600_ ≈ 0.5–0.8 and the cultures were centrifuged at 4000 rpm for 5 min. Pellets were washed with PBST. Pellet was suspended in NADH/NAD extraction buffer (Abcam, Cambridge, UK) with addition of protein inhibitor cocktail (Sigma, St. Louis, MO, USA) and lysed using silica beads in bead-beater for 10 cycles (<60 s) with 1 min intervals. Crude lysates were centrifuged at 10,000 rpm for 10 min at 4 °C and supernatants were collected and filtered using 10 kD spin column to eliminate proteins that can temper with NADH. Filtrates containing NADH as well as NAD^+^ were divided into two reaction sets (1) to measure NAD^+^ (contains both NAD^+^ and NADH) (2) to measure NADH from which NAD^+^ is removed by heating at 60 °C for 30 min to decompose NAD^+^ while NADH remained intact. Afterwards, sample (1) containing NAD^+^ + NADH, sample (2) containing NADH and the standards were transferred to a 96-well plate, this was followed by addition of reaction mix (NAD cycling buffer and NAD cycling enzyme mix, proprietary but just enhances the sensitivity of assay and converts free NAD to NADH). In the end, NADH developer was added and the colorimetric signals was measured at 450 nm using spectrophotometer (Spectramax, Waltham, MA, USA) at 1 h, 2 h, 3 h and 4 h. Individual concentrations of NADH/NAD^+^ were estimated by considering the standard values as reference. Concentation of NAD^+^ was calculated by subtracting concentrations of NAD^+^-NADH that allowed the calculation of the NADH/NAD^+^ ratio [48].

### 4.5. Response of Mtb to Mycobacterial Drugs

The wildtype, RvΔ0687 and C-Rv0687 bacterial strains were grown in 7H9 complete media to an OD_600_ ≈ 0.5–0.8. Final OD_600_ was adjusted to 0.1 (~1 × 10^7^ CFU/mL). To determine the survival of Mtb, the bacterial strains were exposed to Isoniazid (4–0.007 mg/L), Delamanid (4–0.007 mg/L), NMR711 (50–0.390 mg/L), Rifampicin (8–0.062 mg/L) and Bedaquiline (4–0.015 mg/L). The drugs were serially diluted 2-fold in 7H9 media to reach the least concentration gradient. A total of 100 mL of bacteria (~1 × 10^6^ CFU) was added into the wells containing 80 mL of the drug followed by the addition of 20 mL of Alamar Blue to bring the final volume to 200 uL. The plates were incubated at 37 °C for 1–7 days and at the indicated time points, the plates were assessed for bacterial growth and read in a plate reader at OD 590 nm. The survival of bacteria is assessed by plotting the OD 590 nm values on day 7 against drug concentration using GraphPad prism 10 software.

### 4.6. Infection of Bone Marrow Derived Macrophages (BMDMs)

The C57BL6 mice were euthanized, and the femurs and tibias were separated and flushed thoroughly with Dulbecco’s Modified Eagle Medium (DMEM; Invitrogen) to obtain cells. After lysing of red blood cells, the cells were maintained in DMEM, containing 10% FBS and 30% L929 supernatant (complete media) [61]. Cells were cultured in tissue culture flasks for 7 days and seeded at 0.25 × 10^6^ cells per well into 24-well plates. The bacterial strains were grown at 37 °C to OD_600 nm_ ≈ 0.8–1. After washing with phosphate-buffered saline with tyloxapol (PBST), the bacterial suspensions were diluted in DMEM and used to infect BMDMs cells for 3 h at 37 °C in 5% CO_2_ at a multiplicity of infection of 5 (MOI 1:5). Following infection at 3 h, the media were removed, and the wells were washed with PBST before treating with gentamycin for 1 h. Subsequently, the antibiotic media was removed, and the wells were washed with PBST and replenished with fresh DMEM media (complete media). At every time point tested, the infected BMDMs were lysed with 0.01% Triton X-100, and the lysates were serially diluted in PBST and plated on 7H10 agar media to determine the number of colony-forming units (CFUs). The post-infected cell free supernatants were collected before lysis of the cells and analyzed for TNF-α, MIP-1α, IL-1β and IP-10 cytokines.

### 4.7. Measurement of ROS and RNS in BMDMs during Infection

To assess the levels of ROS generated during BMDM infection, 0.1% Nitroblue Tetrazolium Chloride (NBT; Sigma) is added to wildtype, RvΔ0687 and C-Rv0687 infected BMDMs in the 96-well plate and incubated at 37 °C for 30 min. After incubation, the NBT solution was removed, and the cells were suspended in 2 M potassium hydroxide to stabilize the color. Dimethyl Sulfoxide (DMSO) was then added to solubilize the formazan. The amount of formazan, which represents the ROS levels, was quantified by measuring the optical density at 570 nm against DMSO as the reference.

To assess the levels of RNS, post-infected supernatants from wildtype-, RvΔ0687- and C-Rv0687-infected BMDMs were collected, and the assay was performed using the Griess Reagent Kit (Biotium, Fremont, CA, USA) following the guidelines as provided in the kit. Various concentrations of sodium nitrite references were prepared to create a standard curve, which was used to measure the absorbance at 570 nm. Post-infected supernatants were mixed with deionized water and Griess reagent, and the reaction mixture was incubated for ≥30 min at room temperature. The absorbance was then measured at 570 nm, and the optical density reading represented the nitrite concentration in the samples. These values were normalized to the reference values.

### 4.8. Mouse Infection Experiments

The wildtype, RvΔ0687 and C-Rv0687 strains were cultured to an OD_600nm_ ≈ 0.8–1. The bacteria were washed twice with PBST. Subsequently, the samples were sonicated in a Branson cup-horn sonicator, twice for 10 s each time, and then diluted to achieve the desired cell densities of ~1 × 10^6^ CFUs/mL. Female C57BL/6 mice (6–8 weeks old) were obtained from the Jackson Laboratory, and female Rag^−/−^ mice (6–8 weeks old) were obtained from the in-house breeding facility. The mice were infected with a dose of *Mtb* strains (200–250 CFUs per lung) using a Glass-Col chamber as described previously [62] and were euthanized at indicated time points. Lung and spleen were harvested and homogenized with PBST. The CFUs were performed by serially diluting the lysate and plated onto 7H10 plates and colonies were enumerated after 4–6 weeks of incubation at 37 °C and CFUs were estimated per lung and spleen. The animal protocol was approved by the University of Texas at El Paso Institutional Animal Care and Use Committee (IACUC) protocol, A-202004-3 as per NIH principles for animal usage. All the animals were maintained according to the guidelines of IACUC.

## Figures and Tables

**Figure 1 ijms-25-07862-f001:**
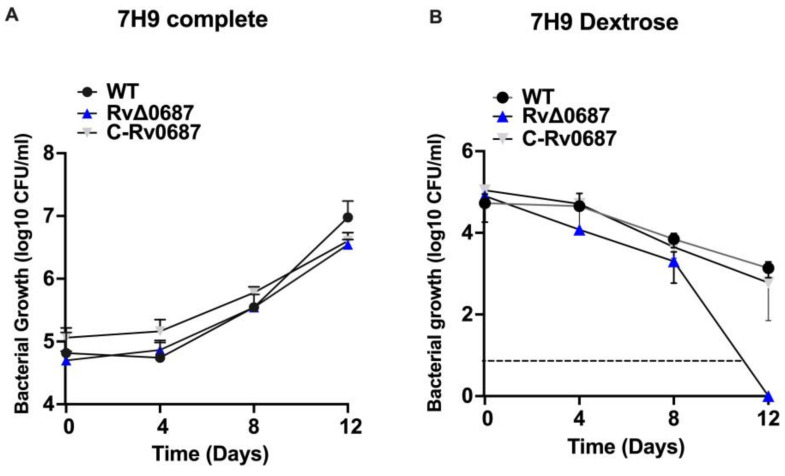
**Growth kinetics of** RvΔ0687. (**A**) **Growth kinetics RvΔ0687 in 7H9 complete media with OADC**. The WT, RvΔ0687 and C-Rv0687 strains were grown in 7H9 complete media, and the growth kinetics were measured at day 0, 4, 8 and 12. Aliquots of bacterial cultures were serially diluted and spotted onto 7H10 complete media and colony forming units (CFUs) were calculated per ml of bacterial culture and graphs were plotted. Experiment was performed in duplicates with biological samples and observed that there was no significance between the *Mtb* strains tested for survival. (**B**) **Growth kinetics RvΔ0687 in 7H9 media with dextrose**. The WT, RvΔ0687 and C-Rv0687 strains were grown in 7H9 dextrose media without OADC until day 12, and the growth was observed by serially diluting the bacterial strains and plating onto 7H10-OADC media. The observed colonies were enumerated to calculate CFUs/mL and graphs were plotted. Experiments were repeated with three biological samples (triplicates), mean ± SD was plotted, and significance was calculated using Two-way ANOVA between the bacterial strains at tested time points.

**Figure 2 ijms-25-07862-f002:**
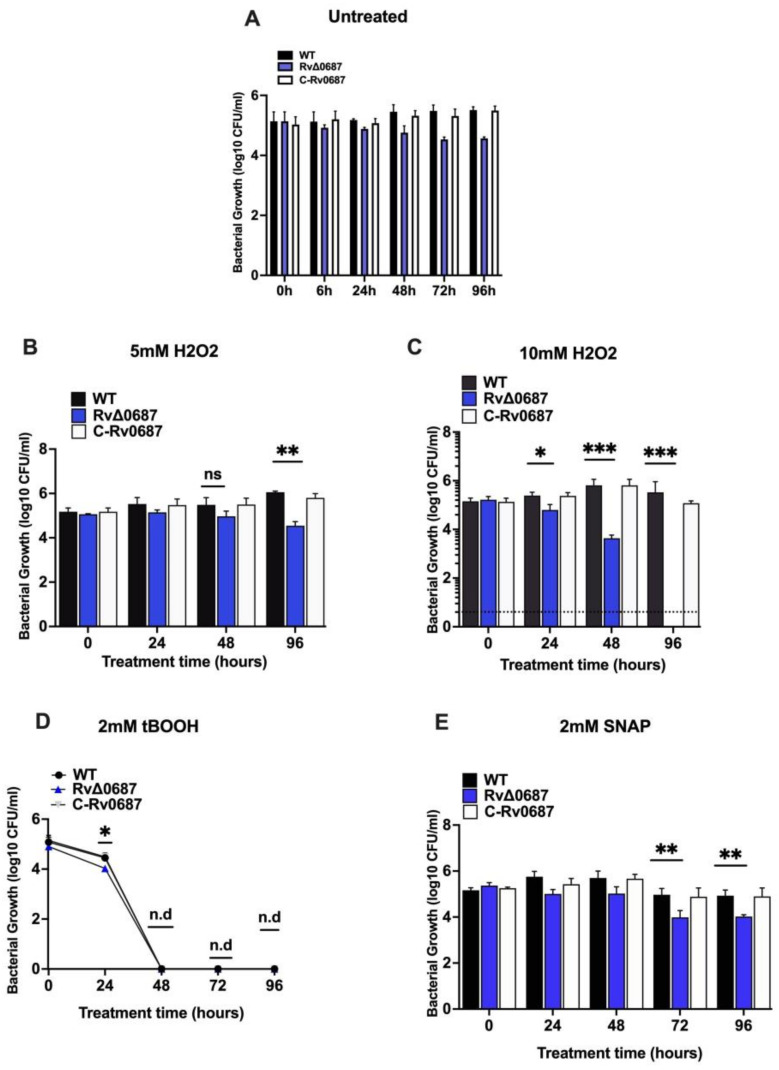
**RvΔ0687 is susceptible to oxidative (ROS) and nitrite stress**. (**A**–**E**) **Susceptibility of RvΔ0687 to oxidative (ROS stress)**. (**A**). **Untreated Strains**. The WT, RvΔ0687 and C-Rv0687 strains were grown in 7H9 complete media, pelleted, washed with PBST, and added to 7H9 dextrose media and the growth was monitored at 0, 6, 24, 48 and 96 h and the CFUs were enumerated by plating onto 7H10 OADC plates. (**B**) **Effect of 5 mM H_2_O_2_ stress**. The WT, RvΔ0687 and C-Rv0687 strains were grown in 7H9 dextrose media and exposed to 5 mM H_2_O_2_ and the growth of all the strains were monitored at 0, 24, 48 and 96 h post-treatment. The cultures were serially diluted at each time point after post-treatment and spotted onto 7H10 OADC plates and CFUs were calculated for 1 mL of bacterial culture. (**C**) **Effect of 10 mM H_2_O_2_ stress**. The WT, RvΔ0687 and C-Rv0687 strains were grown in 7H9 dextrose media and exposed to 10 mM H_2_O_2_ and the growth of all the bacterial strains was monitored. The experiment was carried out in triplicates and significance was calculated using Two-way ANOVA, the error bars indicate the standard deviation (* *p* < 0.1, ** *p* < 0.01, *** *p* < 0.001). Dotted lines represent the limit of detection. (**D**) **Effect of 2 mM tert-butyl hydroxide (tBOOH)**. The WT, RvΔ0687 and C-Rv0687 strains were grown in 7H9 complete media. The pellet was washed twice with PBST and suspended in 7H9 dextrose media, and treated with 2 mM tBOOH and monitored at 0, 24, 48, 72 and 96 h after post-treatment. The sensitivity was analyzed by plating the serially diluted cultures in PBST and spotted onto 7H10. CFUs were enumerated per mL of bacteria and graphs were plotted. The experiment was carried out in triplicates and significance was calculated using Two-way ANOVA and T-test between bacterial strains (* *p* < 0.1) and “n.d” considered as bacteria not detected. (**E**) **Effect of nitrite stress (2 mM SNAP)**. The WT, RvΔ0687 and C-Rv0687 strains were grown in 7H9 dextrose media and treated with 2 mM SNAP, and the bacterial strains were spotted onto 7H10 OADC by serially diluting the cultures. The CFUs were estimated after 3 weeks of incubation and CFUs were calculated per ml of bacterial culture. The experiment was carried out in triplicates and, using Two-way ANOVA, significance was calculated between the bacterial strains (** *p* < 0.01).

**Figure 3 ijms-25-07862-f003:**
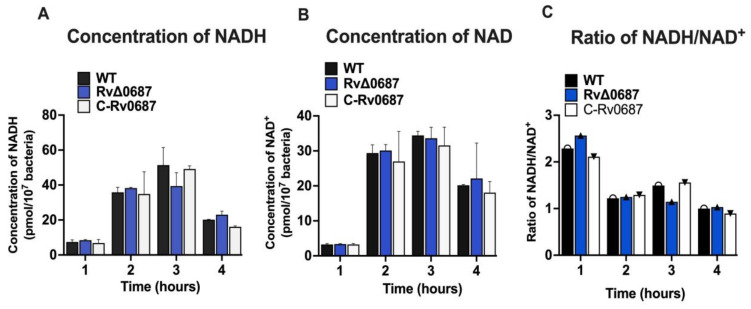
(**A**–**C**) **Rv0687 deletion in *Mtb* does not impact intracellular NADH/NAD^+^ concentrations**.

**Figure 4 ijms-25-07862-f004:**
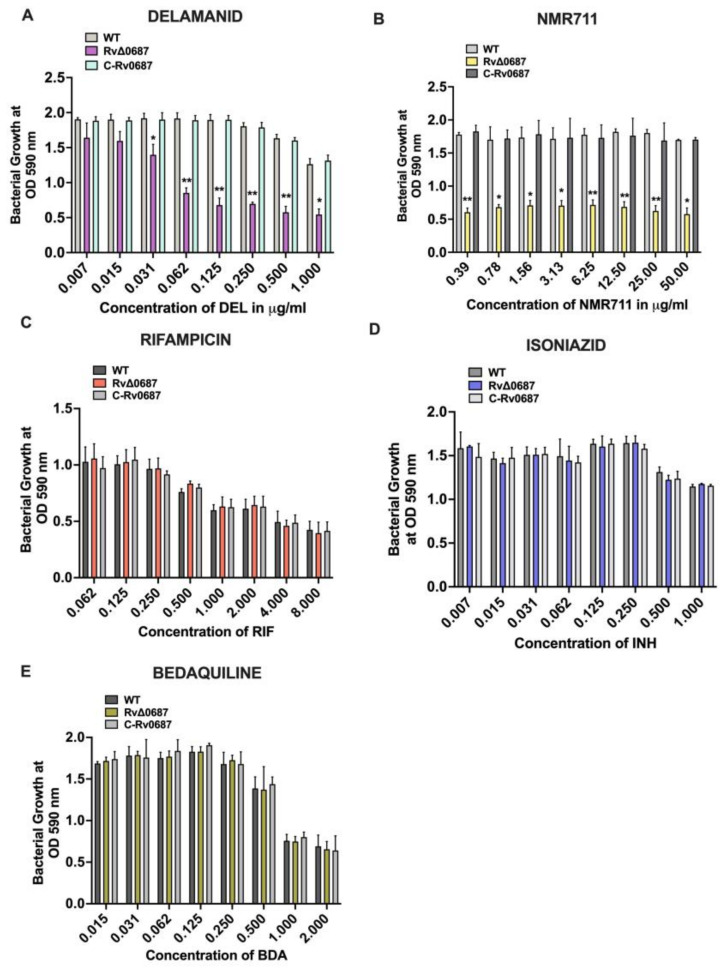
**RvΔ0687 is sensitive to Delamanid and NMR711 treatment**. The WT, RvΔ0687 and C-Rv0687 strains of ~1 × 10^6^ CFU of bacteria were incubated with serially diluted (**A**) **Delamanid**, (**B**) **NMR711**, (**C**) **Rifampicin**, (**D**) **Isoniazid**, (**E**) **Bedaquiline**. Alamar Blue was added and plates were incubated at 37 °C. The absorbance was measured at 590 nm using plate reader from day 1 to day 7 to monitor the bacterial survival and growth. Blank wells containing drugs and Alamar Blue were maintained as controls. Graphs were plotted using OD590 nm values at day 7 and Two-way ANOVA with Tukey’s correction method was used to calculate the significance between WT, RvΔ0687 and C-Rv0687 strains (* *p* < 0.1 and *** p* < 0.01). This experiment was performed in duplicates using appropriate controls.

**Figure 5 ijms-25-07862-f005:**
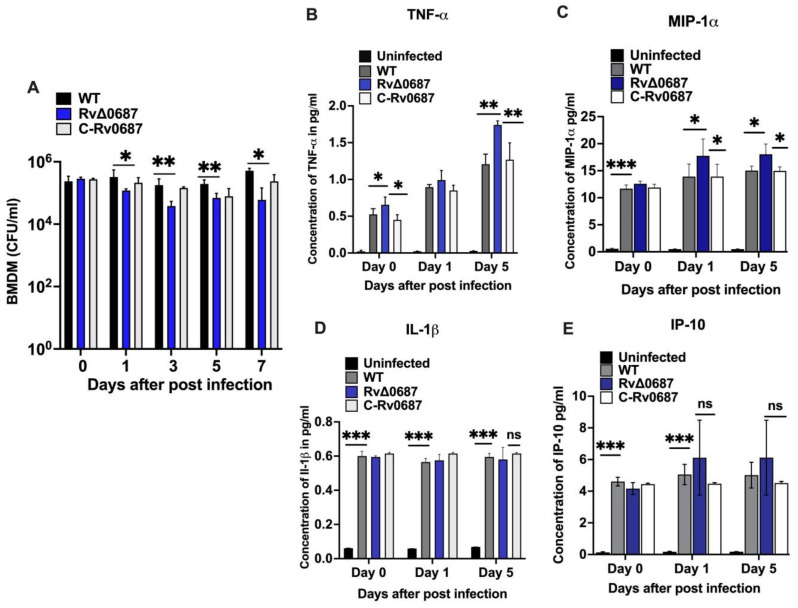
(**A**) **Rv0687 is required for *Mtb* growth and suppression of inflammatory immune response in BMDMs**. The WT, RvΔ0687 and C-Rv0687 strains were used for BMDMs infection at an MOI 1: 5. The in vitro survival was assessed after days 0, 1, 3, 5 and 7 post-infection and bacterial growth was assessed by serially diluting the infected lysate and plated on 7H10 agar and the plates were incubated for 4–6 weeks to calculate the CFUs. The experiment was performed in triplicates and Two-way ANOVA was used to calculate the significance between the bacterial strains. (**B**–**E**) **Cytokine expression during BMDMs infection**. The WT, RvΔ0687 and C-Rv0687 strains infected with BMDMs were analyzed for the expression of TNF-α, IL-1β, MIP-1α and IP-10 cytokines using lysates at day 0, 1 and 5 post-infection. (**B**) TNF-α, (**C**) MIP-1α, (**D**) IL-1β, (**E**) IP-10. The experiment was performed in triplicates and Two-way ANOVA with Tukey’s correction method was used to calculate the significance between uninfected control and infected bacterial strains (*** *p* < 0.001, ** *p* < 0.01 and * *p* < 0.1). “ns” stands for not significant.

**Figure 6 ijms-25-07862-f006:**
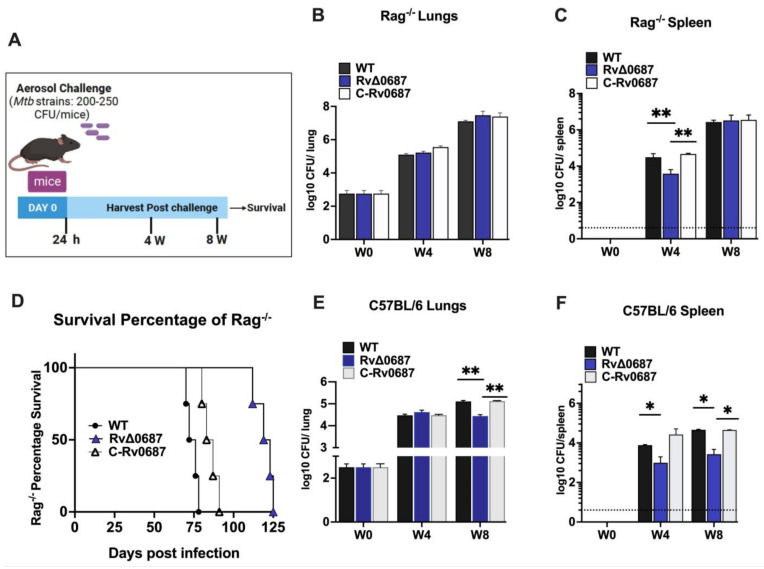
(**A**–**F**) **RvΔ0687 is attenuated for survival in** the **immunocompromised (Rag^−/−^) and immunocompetent (C57BL/6) mice model**. (**A**) **Infection strategy**. (**B**,**C**) **Bacterial load in Rag**^−/−^ **lungs and spleen.** The WT, RvΔ0687 and C-Rv0687 strains were used to infect the Rag^−/−^ mice (n = 4 animals per group) and euthanized at W0, W4 and W8. Lungs and spleen were harvested, homogenized, and used to enumerate bacterial load via CFU assay. The experiment was performed with n = 5/group and significance was calculated using Two-way ANOVA ** *p* < 0.01. (**D**) **Survival analysis of WT and RvΔ0687 in Rag**^−/−^. The survival of mice post infection with WT, C-Rv0687 and RvΔ0687 was performed with n = 4 animals per group, and the survival was monitored for 125 days and plotted using the Log rank test. (**E**,**F**) **Bacterial load in C57BL/6 lungs and spleen**. The WT, RvΔ0687 and C-Rv0687 strains were used to infect C57BL/6 mice (n = 4 animals per group) and were euthanized at W0, W4 and W8. Lungs and spleen were processed for bacterial enumeration and CFUs were enumerated per lung and spleen. Dotted line in (**C**,**F**) represents the limit of detection. The experiment was performed with appropriate statistical numbers and significance was calculated using Two-way ANOVA (** *p* < 0.01 and * *p* < 0.1).

## Data Availability

Data are contained within the article and Appendix A.

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
