# Peer review of "Rv0687 a Putative Short-Chain Dehydrogenase Is Required for In Vitro and In Vivo Survival of Mycobacterium tuberculosis"

_ijms, 2024, doi:10.3390/ijms25147862_

Round 1

Reviewer 1 Report

Comments and Suggestions for Authors

In this manuscript entitled “Rv0687 a Putative Short-Chain Dehydrogenase is indispensable for pathogenesis of Mycobacterium tuberculosis”, Bjargavi et al. Describe the role of RV 0687, a part of the short-chain dehydrogenase/reductase family, in the virulence of Mycobacterium tuberculosis. To do so, the author engineered a deleterious Rv0687 strain and its complemented mutant and tested their survival in vitro when exposed to RONS (reactive oxygen and nitrogen species) and to different antitubercular drugs. The authors also tested the pro-inflammatory response of Rv0687 in BMDM macrophages following infection. The role of Rv0687 in M. tb pathogenesis was determined in vivo in RAG -/- and BL/6 mice. I have significant concerns regarding the method description. In my opinion, as it is, the methods don’t allow the reader to repeat the experiments described in the manuscript due to discrepancies between the method section and the main manuscript or the lack of information on the method or conditions (e.g. multiple names to define the media used, especially the 7H9 media with dextrose-see line 126-127 in comparison to the associated figure) used. Also, I would like to point out that the research hypothesis is based partly on unpublished data from an RNA-seq experiment performed by the authors. I suggest the authors reconsider the publication of the RNA-seq data or support the hypothesis with data that have already been published. You will find below my main comments. However, a thorough revision of this manuscript is required in the methods section and throughout the main manuscript.

1- line 150-151: Can the authors explain why they chose these specific concentrations of ions for testing?

2-Figure 2: Figure 2A shows that Rv∆0687 has a lower CFU count at 0 and 24h post-treatment with 5mM H2O2 compared to the WT and C-Rv0687. This trend is not observable when the deleterious mutant is treated with 10 mM H2O2. These results could be explained by the difference in the number of CFUs used at the beginning of the experiment (inoculum), which can be falsely interpreted as a reduction in CFU due to RNOS. The authors need to add a non-treated group for each time point and each strain tested for all RNOS tested.

3- Figure 3: According to the method (lines 499-502), the authors mentioned that the NAD+/NADH ratio was performed in 7H9 complete media. Can the authors explain why they chose this media instead of 7H9-dextrose? Also, according to the available data, is it known if Rv0687 is constitutively expressed in M.tb in vitro and if the growth conditions may alter the expression of this gene?

4- Figure 4: The authors need to revise the units of the y-axis of the graphs. In the legend, it is mentioned that the fluorescence readings are at 560nm and 590 nm. To my knowledge, the values on the y-axis are not fluorescence values. Does the y-axis represent the ratio of 560nm and 590nm? This needs to be clarified.

5- Figure 5A: At 5 and 7 days post-infection, we can observe that C-Rv0687 is not fully restoring the WT phenotype (at 1 and 3 days post-infection, the complemented strain has a similar bacterial burden to the WT). Can the authors explain or add a comment in the manuscript about the statistical significance of these results?

6- Line 262: changed for in vivo

7- Figure 6: The authors should report the CFU± standard deviation in a Table. Also, for Figure 6C and F at W0, the authors seem to report 0 CFU (this is why a Table would be helpful as it won’t rely on the reader’s interpretation). However, what is the CFU detection limit for this experiment? It is unlikely that 0 CFU was observed. If the authors didn’t observe any colonies at the lowest dilution plated, they should mention” below the limit of detection,” if it was not tested, they should report “not determined.”

8-Line 335: Did the authors use polysorbate (Tween)? As mentioned in the method, they used tyloxapol. Please clarify.

9- Lines 365-367: Is a homolog of Rv0687 described in several mycobacteria or in one mycobacterium species? This sentence is confusing as the author described only the homolog in MAP in the following sentence. Please add references and clarify.

10- Lines 296-398: This sentence is confusing as the manuscript doesn’t mention Rv0148 and its association with antibiotic susceptibility before this sentence.

11- Lines 401-403: Please explain or reference this statement.

12-Line 467: The authors should mention that they use M.tb CDC1551 in the method. It is reported in the supplementary material, but adding it to the main manuscript is relevant as the Rv gene numbering is used (potential confusion with the H37Rv strain).

13-Line 474: Please revise all drug concentration units in the manuscript. The unit should be ug/ml or mg/L, not mg/mL.

14-Lines 482 and 486: Fig S5 should be reported instead of S1.

15-Lines 482-483: The authors must describe or reference the protocol used for gDNA extraction, library preparation, WGS and data analysis. If Illumina provided the protocol (to my knowledge, Illumina only recommends kits for library preparation and WGS. However, several of these kits could be used with modifications), the authors must reference or explain this protocol.

16- Line 490-491: In which media were the cells grown, and were the bacteria washed to remove any catalase if 7H9 complete media was used? This protocol is incomplete.

17- Line 544: what is the concentration of Triton used for lysis?

18- Line 570: Did they use male or female mice for the experiment? 

Comments on the Quality of English Language

I advise to revise the manuscript for spellcheck.

Author Response

In this manuscript entitled “Rv0687 a Putative Short-Chain Dehydrogenase is indispensable for pathogenesis of Mycobacterium tuberculosis”, Bhargavi et al. Describe the role of RV 0687, a part of the short-chain dehydrogenase/reductase family, in the virulence of Mycobacterium tuberculosis. To do so, the author engineered a deleterious Rv0687 strain and its complemented mutant and tested their survival in vitro when exposed to RONS (reactive oxygen and nitrogen species) and to different antitubercular drugs. The authors also tested the pro-inflammatory response of Rv0687 in BMDM macrophages following infection. The role of Rv0687 in M. tb pathogenesis was determined in vivo in RAG -/- and BL/6 mice. I have significant concerns regarding the method description. In my opinion, as it is, the methods don’t allow the reader to repeat the experiments described in the manuscript due to discrepancies between the method section and the main manuscript or the lack of information on the method or conditions (e.g. multiple names to define the media used, especially the 7H9 media with dextrose-see line 126-127 in comparison to the associated figure) used. Also, I would like to point out that the research hypothesis is based partly on unpublished data from an RNA-seq experiment performed by the authors. I suggest the authors reconsider the publication of the RNA-seq data or support the hypothesis with data that have already been published. You will find below my main comments. However, a thorough revision of this manuscript is required in the methods section and throughout the main manuscript.

Response. We thank the reviewer for summarizing our study and the valuable inputs provided. We have thoroughly revised the manuscript and modifications have been highlighted in yellow in the revised manuscript.

1- line 150-151: Can the authors explain why they chose these specific concentrations of ions for testing? 

Response. Thank you. We chose these concentrations based on published literature on Mtb knockout mutants showing phenotypes in response to metal ion concentrations. Specifically, we tested 0.01% and 0.05% concentrations for FeSO4, 50 and 75 μM concentrations for Zinc and Copper (PMID: 23562345 and PMID: 35745538)

2-Figure 2: Figure 2A shows that Rv∆0687 has a lower CFU count at 0 and 24h post-treatment with 5mM H2O2 compared to the WT and C-Rv0687. This trend is not observable when the deleterious mutant is treated with 10 mM H2O2. These results could be explained by the difference in the number of CFUs used at the beginning of the experiment (inoculum), which can be falsely interpreted as a reduction in CFU due to RNOS. The authors need to add a non-treated group for each time point and each strain tested for all RNOS tested. 

Response. Thank you. We apologize for our negligence, the 5mM H2O2 graph we initially uploaded is incorrect. The exact graph for Fig.2A has been added in the revised manuscript. We used an initial inoculum of 10^5 CFUs for all experiments in this study. Additionally, we included non-treated group for each time point as part of our experimental design, which has now been added in Fig.2 A in the revised manuscript.

3- Figure 3: According to the method (lines 499-502), the authors mentioned that the NAD+/NADH ratio was performed in 7H9 complete media. Can the authors explain why they chose this media instead of 7H9-dextrose? Also, according to the available data, is it known if Rv0687 is constitutively expressed in M. tb in vitro and if the growth conditions may alter the expression of this gene? 

Response. Thank you for your question. This experiment aimed to determine the impact of Rv0687 deletion on cell homeostasis by examining the NADH/NAD ratio. We chose to perform this assay in 7H9 complete media with OADC because 7H9 dextrose media, lacks catalase, and thus can impact desired measurements of cell homeostasis. As per available gene expression datasets Rv0687 is reported as non-essential gene (PMID: 28096490).. As per TB database, its reported that its expression increases in repose to exposure to pyrazinamide, econazole (TB database) while its expression decreases during hypoxia. We also found in our studies that exposure of H37Rv to antimycobacterial compound NMR 711 (PMID: 34194673) increases expression of Rv0687.

4- Figure 4: The authors need to revise the units of the y-axis of the graphs. In the legend, it is mentioned that the fluorescence readings are at 560nm and 590 nm. To my knowledge, the values on the y-axis are not fluorescence values. Does the y-axis represent the ratio of 560nm and 590nm? This needs to be clarified.

Response. Thank you for bringing this to our attention. We apologize for the confusion. The values on the Y-axis represent the OD readings at 590nm, not the fluorescence values as indicated initially. We have revised the figure legends accordingly to clarify this. The Y-axis in Fig.4 has been updated accordingly.

5- Figure 5A: At 5- and 7-days post-infection, we can observe that C-Rv0687 is not fully restoring the WT phenotype (at 1- and 3-days post-infection, the complemented strain has a similar bacterial burden to the WT). Can the authors explain or add a comment in the manuscript about the statistical significance of these results? 

Response. Thank you. We acknowledge the slight change in survival of C-Rv0687 at 5- and 7-days post-infection, which is not statistically significant. This could potentially be attributed to increased hypoxia conditions and the use of mycobacterial vector pMV361 for complementing the Rv0687 gene which has hsp60 promoter (sensitive to hypoxia). We have included this explanation in the revised manuscript line no. 236-238 and 429-431.

6- Line 262: changed for in vivo

Response. The sentence is modified now to in-vivo in the revised manuscript line no. 269.

7- Figure 6: The authors should report the CFU± standard deviation in a Table. Also, for Figure 6C and F at W0, the authors seem to report 0 CFU (this is why a Table would be helpful as it won’t rely on the reader’s interpretation). However, what is the CFU detection limit for this experiment? It is unlikely that 0 CFU was observed. If the authors didn’t observe any colonies at the lowest dilution plated, they should mention” below the limit of detection,” if it was not tested, they should report “not determined.”

Response. We appreciate the suggestion provided by the reviewer. In FIG.6 C and 6F, of the revised manuscript, we have adjusted the graphs to reflect the limit of detection for CFU counts. We believe that representing the CFUs in graphical format is more appropriate for enumerating bacteria in the lungs and spleen compared to a table. Therefore, we would like to maintain the current representation with modifications incorporated.

8-Line 335: Did the authors use polysorbate (Tween)? As mentioned in the method, they used tyloxapol. Please clarify.

Response. Thank you. We used tyloxapol in the current study. We have modified the sentence in line no. 344 accordingly

9- Lines 365-367: Is a homolog of Rv0687 described in several mycobacteria or in one mycobacterium species? This sentence is confusing as the author described only the homolog in MAP in the following sentence. Please add references and clarify. 

Response. Thank you. The sentence is modified now in the revised manuscript line no. 375-376 to “Interestingly, a homolog of Rv0687 in other mycobacterial species is characterized as a mycofactocin-associated dehydrogenase with non-exchangeable NAD-cofactors PMID: 33414309.

10- Lines 296-398: This sentence is confusing as the manuscript doesn’t mention Rv0148 and its association with antibiotic susceptibility before this sentence. 

Response. Thank you. The sentence has been modified now with addition of references in the revised manuscript line no. 405-408.

11- Lines 401-403: Please explain or reference this statement.

Response. Thank you. The reference for this statement is added in the revised manuscript line no. 412

12-Line 467: The authors should mention that they use M. tb CDC1551 in the method. It is reported in the supplementary material, but adding it to the main manuscript is relevant as the Rv gene numbering is used (potential confusion with the H37Rv strain). 

Response. Thank you. The following modification has been added to the revised manuscript line no. 494

13-Line 474: Please revise all drug concentration units in the manuscript. The unit should be ug/ml or mg/L, not mg/mL. 

Response. Thank you for the suggestion. We have modified the drug concentration units to mg/L wherever required throughout the manuscript

14-Lines 482 and 486: Fig S5 should be reported instead of S1.

Response. Thanks for the suggestion. The supplementary figure text is modified in line no.499 and 500.

15-Lines 482-483: The authors must describe or reference the protocol used for gDNA extraction, library preparation, WGS and data analysis. If Illumina provided the protocol (to my knowledge, Illumina only recommends kits for library preparation and WGS. However, several of these kits could be used with modifications), the authors must reference or explain this protocol.

Response.  Thank you. The references for the methods has been updated in the revised manuscript materials and methods 4.2 section

16- Line 490-491: In which media were the cells grown, and were the bacteria washed to remove any catalase if 7H9 complete media was used? This protocol is incomplete. 

Response. Thank you. The following sentence has been added in the revised manuscript line no. 508-510

17- Line 544: what is the concentration of Triton used for lysis?

Response. Thank you 0.01% concentration of TritonX-100 is used for lysing the cells. The concentration is added in the revised manuscript line no. 562-563. 

18- Line 570: Did they use male or female mice for the experiment? 

Response. We used female Rag-/- and female C57BL/6 mice. The following information has been added in the revised manuscript line no. 588 and 589.

Reviewer 2 Report

Comments and Suggestions for Authors

Accepted with major revision

Question #1:  It would be better for authors to further investigate other possible functions of Rv0687 within the bacterium to better understand its role in M. tuberculosis infections. Furthermore, the present study did not explore potential differences in immune responses between different strains of mice infected with the Rv0687 deletion mutant versus wildtype M. tuberculosis.

Question #2: The authors need to revise the article to reduce the similarity index, which is currently too high at 70%.

Question #3: What are the consequences of Rv0687 deletions on antibiotic susceptibility and the host immune response during M. tuberculosis infection? How does Rv0687 influence cytokine secretion and Bacterial load.

Good Luck

Comments on the Quality of English Language

Minor editing of English language required

Author Response

We thank the reviewer for summarizing our study and the valuable suggestions provided.

Question #1:  It would be better for authors to further investigate other possible functions of Rv0687 within the bacterium to better understand its role in M. tuberculosis infections. Furthermore, the present study did not explore potential differences in immune responses between different strains of mice infected with the Rv0687 deletion mutant versus wildtype M. tuberculosis.

Response. We thank the reviewer for raising this question. While we have not studied immune responses in the current study, we recognize its importance and have planned to address this aspect in our ongoing and future studies.

Question #2: The authors need to revise the article to reduce the similarity index, which is currently too high at 70%.

Response. We agree with reviewer and have thoroughly revised the manuscript to significantly reduce the similarity index, but it might be due to our preprint.

Question #3: What are the consequences of Rv0687 deletions on antibiotic susceptibility and the host immune response during M. tuberculosis infection? How does Rv0687 influence cytokine secretion and Bacterial load.

Response. We thank reviewer for this question and have incorporated this explanation in discussion. Rv0687 is a short chain dehydrogenase (SDR) and SDRs have been shown to play role in managing peroxidase stress, detoxification, in-vitro, in-vivo survival, drug resistance and overall homeostasis in various organisms including bacteria. They can serve as oxidoreductases and bind to NADH to donate electrons in the various reactions required for bacterial physiology. Deletion of Rv0687 is causing enhanced antibiotic susceptibility might be due to perturbation of these metabolic reactions.

As deletion of Rv0687 perturb Mtb physiology, it makes deletion strain to be more attenuated than WT strain leading to enhanced proinflammatory cytokine secretion and control by immune response leading to decreased bacterial burden.

Reviewer 3 Report

Comments and Suggestions for Authors

This study describes the functional role of Rv0687, a member of SDRs family, in Mtb pathogenesis. They constructed the Rv0876-deleted mutant and complemented strain and then compared their growth rate in 7H9 media with/without OADCs containing ROS or RNI-related materials, including H2O2, nitrite, and so on, and a change of drug susceptibility. The change in their virulence was determined by the intracellular growth rate in macrophages and the growth rate in immunocompromised and immunocompetent mice. Rv0687-deleted Mtb strain could not survive in 7H9 dextrose without ADC, was susceptible in H2O2 and tBOOT, and increased susceptibility to antimycobacterial drug delamanid, and attenuated its virulence in macrophages and mice. This paper provides valuable information on the role of a member of the SDR family. 

 1. In the title, I cannot agree that Rv0678 is indispensable for TB pathogenesis. Based on Fig. 5 and Fig. 6, the deletion of Rv0678 led to a slight attenuation of Mtb virulence, and the role of Rv0678 is a redundancy family.

2.  Figure S4 should be moved to the main figure or described next to Fig. 2. 

3. The sentence between Line 114 and Line 115 is not suitable for this place. Please move to it next to the description of Fig S4.

4. In Figure 5A, the Y-axis scale should be changed from log scale to numerical. Because it is too hard to find significant differences intuitively. 

5.    Figure 5A  Fig. 5. 

6. In Fig. 5, the main killing mechanism of Mtb in macrophages is phagolysosomal fusion. ROS also activates this pathway. Is there any difference in phagosome-lysosome fusion between mutant and WT Mtb strains?  

7.  In Figure 6, the results are inconsistent. For example, the mutant load in the tissue is depressed only in the spleen at 4 weeks in Rag-/- mice, and in the lung at 8 weeks and in the spleen at 4 weeks and 8 weeks in the C57BL/6 mice. I think that the mouse number is too small per group (This study used 4 mice per group).

8.   In Figure 6, please give the pathologic findings (ex, HE stain)

9.  In line 262, ‘in vitro’ should be changed to ‘in vivo’.

10. The part of '5. future studies' must be deleted and, instead, included shortly into Discussion section.

Author Response

This study describes the functional role of Rv0687, a member of SDRs family, in Mtb pathogenesis. They constructed the Rv0876-deleted mutant and complemented strain and then compared their growth rate in 7H9 media with/without OADCs containing ROS or RNI-related materials, including H2O2, nitrite, and so on, and a change of drug susceptibility. The change in their virulence was determined by the intracellular growth rate in macrophages and the growth rate in immunocompromised and immunocompetent mice. Rv0687-deleted Mtb strain could not survive in 7H9 dextrose without ADC, was susceptible in H2O2 and tBOOT, and increased susceptibility to antimycobacterial drug delamanid, and attenuated its virulence in macrophages and mice. This paper provides valuable information on the role of a member of the SDR family. 

Response. We thank the reviewer for summarizing our study and the valuable suggestions provided. We have thoroughly revised the manuscript to address the comments. The modifications have been highlighted in yellow in the revised manuscript.

  1. In the title, I cannot agree that Rv0678 is indispensable for TB pathogenesis. Based on Fig. 5 and Fig. 6, the deletion of Rv0678 led to a slight attenuation of Mtb virulence, and the role of Rv0678 is a redundancy family.

Response. Thank you for your feedback. Based on your suggestion, we have modified the title of manuscript to “Rv0687 a Putative Short-Chain Dehydrogenase is required for in-vitro and in-vivo survival of Mycobacterium tuberculosis”.

  1. Figure S4 should be moved to the main figure or described next to Fig. 2. 

Response. We thank the reviewer for the suggestion we moved supplementary figure S4 to main text which is currently Fig.S2 line no. 136 in the revised manuscript.

  1. The sentence between Line 114 and Line 115 is not suitable for this place. Please move to it next to the description of Fig S4.

Response. Thank you for your suggestion. The sentence between lines 114 and 115 has been moved to the discussion section in the revised manuscript at line no. 349-350

  1. In Figure 5A, the Y-axis scale should be changed from log scale to numerical. Because it is too hard to find significant differences intuitively. 

Response. Thank you for your suggestion. In fig. 5A, the Y-axis scale has been changed from log scale to numerical scale to facilitate easier identification of significant differences. This modification has been made in the revised manuscript

  1. Figure 5A àFig. 5. 

  1. In Fig. 5, the main killing mechanism of Mtb in macrophages is phagolysosomal fusion. ROS also activates this pathway. Is there any difference in phagosome-lysosome fusion between mutant and WT Mtb strains?  

Response. We thank the reviewer for raising this interesting question. While we have not studied phagosome lysosome fusion in the current study, it is an important aspect that will be the focus of our ongoing and future studies.

  1. In Figure 6, the results are inconsistent. For example, the mutant load in the tissue is depressed only in the spleen at 4 weeks in Rag-/- mice, and in the lung at 8 weeks and in the spleen at 4 weeks and 8 weeks in the C57BL/6 mice. I think that the mouse number is too small per group (This study used 4 mice per group).

Response. Thank you for your observation. In our study, we initially used 5 mice/group for infection and survival experiments. However, due to the mortality in the survival studies,

we reported results within n=4/group. All post-infection studies were performed with n=5/group, which is considered an adequate sample size to calculate statistical significance.

The variation in bacterial load between the lungs vs spleen may be attributed to the dissemination of Mtb infection from the lungs to spleen during the early stages of infection, and the complex interplay between Mtb and the host immune response. This is discussed  in lines 453-457 and 474-475 of the revised manuscript.

  1.  In Figure 6, please give the pathologic findings (ex, HE stain)

Response. Thank you for this suggestion. In the current study, we have not performed any immunohistochemistry staining including H&E staining. We will include these analyses in our future studies to provide more comprehensive pathological findings.

  1. In line 262, ‘in vitro’ should be changed to ‘in vivo’.

Response. Thank you for pointing this out. We have added corrected "in vitro" to "in vivo" in the revised manuscript, now reflected in line 273.

  1. The part of '5. future studies' must be deleted and, instead, included shortly into Discussion section.

Response. Thank you for your suggestion. We have removed the Future studies section and integrated the relevant points into the discussion section line no. 476-479 of the revised manuscript.

Round 2

Reviewer 1 Report

Comments and Suggestions for Authors

The authors have answered the reviewer's questions/comments and improved the manuscript in the revised version. However, I have a few comments for the authors prior to publication. 

1-The authors must revise the manuscript for sentence structure and spellcheck, especially on the modified sentences. Some of the sentences are hard to understand. I also added a few of the sentences that were not modified but are hard to understand below as examples.

2- New title: Please verify the new title as it contains twice in-vitro. Do the authors mean "in-vitro and in-vivo"?

3-  line 43: via instead of vis?

4- Lines 64-67: Please revise this part of the sentence: "which aid in managing peroxidase stress, detoxification, in-vitro, in-vivo survival, drug resistance and overall homeostasis". I think the authors mean "detoxification in vitro"

5-line 98: please revise the sentence

6- Figure 2A: Please verify the y-axis unit.

7- lines 212-213: It is the first time in the revised manuscript that the authors mentioned the compound NMR711. I think a reference should be added, or the authors should provide information about this compound. 

8- line 331: The term "polysorbate" is used and should be changed to tyloxapol, according to the authors' response to the reviewers' comments. 

9- line 402-403: Please revise this sentence. I don't understand the meaning of this sentence.

Comments on the Quality of English Language

The authors must revise the manuscript for sentence structure and spellcheck, especially on the modified sentences. Some of the sentences are hard to understand. 

Author Response

1-The authors must revise the manuscript for sentence structure and spellcheck, especially on the modified sentences. Some of the sentences are hard to understand. I also added a few of the sentences that were not modified but are hard to understand below as examples.

Response. We thank the reviewer for the valuable inputs and suggestions. We have thoroughly revised the manuscript for sentence structure and spelling, especially focusing on the modified sentences. Additionally, we have reviewed and clarified the sentences mentioned by the reviewer. All modifications have been highlighted in green in the revised manuscript.

2- New title: Please verify the new title as it contains twice in-vitro. Do the authors mean "in-vitro and in-vivo"?

Response. Thank you for pointing that out. We have corrected the title now in the revised manuscript to “Rv0687 a Putative Short-Chain Dehydrogenase is required for in-vitro and in-vivo survival of Mycobacterium tuberculosis’’.

3-  line 43: via instead of vis?

Response. Thank you. We have corrected the word from ’’vis’’ to ‘via’, on line no. 42 in the revised manuscript.

4- Lines 64-67: Please revise this part of the sentence: "which aid in managing peroxidase stress, detoxification, in-vitro, in-vivo survival, drug resistance and overall homeostasis". I think the authors mean "detoxification in vitro"

Response. Thank you for the suggestion. We have revised the sentence to clarify in line no. 66 of the revised manuscript.

5-line 98: please revise the sentence

Response. Thank you. We have revised the sentence, and the changes are reflected in lines 96-99 of the revised manuscript

6- Figure 2A: Please verify the y-axis unit.

Response. Thank you for bringing this to our attention. We have modified the Y-axis unit in Fig.2A in the revised manuscript.

7- lines 212-213: It is the first time in the revised manuscript that the authors mentioned the compound NMR711. I think a reference should be added, or the authors should provide information about this compound. 

Response. Thank you for your suggestion. We have added information about the compound NMR711and included a reference in lines 211-213 of the revised manuscript

8- line 331: The term "polysorbate" is used and should be changed to tyloxapol, according to the authors' response to the reviewers' comments. 

Response. Thank you for pointing this out. We have changed polysorbate to “tyloxapol” in line no. 336 of the revised manuscript.

9- line 402-403: Please revise this sentence. I don't understand the meaning of this sentence.

Response. Thank you for your feedback. We have revised the sentence for clarity, and the changes are reflected in lines 400-404 of the revised manuscript.

Reviewer 3 Report

Comments and Suggestions for Authors

By my previuos comment 8, I think the pathologic finding should be needed, especially in the case of this study, please describe this point in the discussion section. 

Author Response

By my previuos comment 8, I think the pathologic finding should be needed, especially in the case of this study, please describe this point in the discussion section. 

Response. Thank you for your valuable feedback. We agree that pathological findings will add to this study. As the CFU of WT and RvΔ0687 were similar in the lungs at 4 weeks (an initial time point) we do not expect tremendous differences in pathology of lungs by H&E staining, however these can be analyzed in our future studies. Accordingly, we have incorporated this point into the discussion section. Please see the revised manuscript on lines 468-471, which have been highlighted in green.
